# MicroRNA-371a-3p Represents a Novel and Effective Diagnostic Marker for Testicular Germ Cell Tumours: A Real-World Prospective Comparison with Conventional Approaches

**DOI:** 10.3390/pathophysiology32040054

**Published:** 2025-10-06

**Authors:** Margherita Palermo, Carolina D’Elia, Giovanni Mazzucato, Christine Mian, Christine Schwienbacher, Esther Hanspeter, Silvia Clauser, Salvatore Mario Palermo, Armin Pycha, Isabel Heidegger, Igor Tsaur, Emanuela Trenti

**Affiliations:** 1Department of Urology, Eberhard-Karls-University of Tübingen, 72076 Tübingen, Germany; margherita.palermo@med.uni-tuebingen.de (M.P.); igor.tsaur@med.uni-tuebingen.de (I.T.); 2Department of Urology, Central Hospital of Bolzano, 39100 Bolzano, Italysilvia.clauser@sabes.it (S.C.); salvatore.palermo@sabes.it (S.M.P.); armin.pycha@sabes.it (A.P.); 3Department of Pathology, Central Hospital of Bolzano, 39100 Bolzano, Italy; christine.mian@sabes.it (C.M.); christine.schwienbacher@sabes.it (C.S.); esther.hanspeter@sabes.it (E.H.); 4Medical School, Sigmund Freud Private University, 1020 Vienna, Austria; 5Department of Urology, Medical University of Innsbruck, 6020 Innsbruck, Austria; isabel-maria.heidegger@i-med.ac.at

**Keywords:** testicular germ-cell tumour, MiRNA, tumour markers, small testicular masses, seminoma, non-seminoma, real-life study

## Abstract

**Background/Objectives:** Testicular germ cell tumours (GCT) have high cure rates, especially in early stages. MicroRNA-371a-3p (M371) has recently emerged as a highly sensitive biomarker for malignant GCTs, except teratoma. This study aimed to evaluate the diagnostic performance of M371-test in a real-life clinical setting, compared to conventional markers alpha-fetoprotein (AFP), lactate-dehydrogenase (LDH), and beta-human chorionic gonadotropin (β-HCG) in patients with suspected GCT. **Methods:** The study, approved by the Ethic-Committee of the Provincial Hospital of Bolzano (N.97-2021), included 91 M371-tests, performed from March 2021 to May 2025. A total of 75 patients had suspected GCT; 19 healthy males served as control. Serum levels of M371, AFP, LDH, and β-HCG were compared with final histopathological diagnosis. M371 was also assessed in controls to evaluate test performance. Secondary analyses investigated correlations between preoperative M371 levels and tumour size in non-metastatic patients, and between M371-levels and clinical stage in the entire GCT cohort. A cut-off of RQ > 5 (relative quantification) was used to calculate sensitivity, specificity, and predictive values. **Results:** M371 showed a sensitivity of 90.9% and specificity of 89.3%, outperforming in terms of sensitivity AFP (20.4%/96.4%), LDH (40.9%/96.4%), and β-HCG (43.1%/100%). Positive predictive value (PPV) and negative predictive value (NPV) were 93.0% and 86.2%, respectively. Sensitivity was 95% for non-seminomas and 87.5% for seminomas. In non-metastatic patients, M371 levels correlated with tumour size and were significantly higher in advanced stages (median RQ 1128.35 vs. 98.36; *p* = 0.015). **Conclusions:** M371 showed excellent diagnostic performance, even for small tumours, supporting its clinical use. Further studies are needed to define its role in treatment planning and follow-up.

## 1. Introduction

Testicular cancer is an uncommon tumour, accounting for 1.7% of all male malignancies in adults, with almost 72.000 estimated new cases worldwide in 2022 [1]. Despite its overall low incidence, it is the most common tumour in men aged between 15 and 44 years in Europe; moreover, it is expected an overall increasing number of cases by 2035 [2]. Histologically, 95% of these tumours are germ cell tumours (GCTs), further divided into 55–60% pure seminomas and 40–45% non-seminomas [3]. Their survival rate is very high, near to 95% in the clinical stage I and II and 75% in the clinical stage III [4,5]. Approximately 85% of seminomas and 60% of non-seminomas are diagnosed in clinical stage I [6]. Serum tumour markers alpha-fetoprotein (AFP), beta-human chorionic gonadotropin (β-HCG), and lactate-dehydrogenase (LDH) play an important role in diagnosis and management of GCT. Their levels should be measured both before and after orchiectomy, due to their association with histological type and their prognostic value. Due to their clinical relevance, these markers were first included in the fifth edition of TNM classification and have since then contributed significantly to diagnosis, response assessment, relapse detection, and risk stratification of GCTs [7].

AFP is an oncofetal protein, mostly produced by yolk sac during pregnancy, gastrointestinal tract, and liver; for this reason, it does not rise by pure seminomas and trophoblastic tumours, while it can be high in most yolk sac tumours and in some embryonal carcinomas, but also in the case of liver diseases or hereditary factors and in weekend drinkers. Its half-life in serum is from 5 to 7 days [7,8].

β-HCG is produced by the syncytiotrophoblast of the placenta during pregnancy. By choriocarcinomas, its level is always increased, while it could be elevated in almost 50% of embryonal carcinomas and in one third of seminomas. However, β-HCG can also be secreted by other malignancies and in individuals with chronic marijuana abusus. Its half-life in serum is from 18 to 36 h [8,9,10].

LDH is a cellular enzyme with several subtypes, which rises in all cases of cell lysis, and thus not only in the case of malignancies but also in the case of myocardial infarction or of liver and kidney diseases; for this reason, it is a very unspecific marker, but it can have a prognostic value if it is very high. Its serum half-life is approximately 24 h [8].

Unfortunately, due to their low sensitivity—approximately 60% in non-seminomas and 20% in seminomas—the utility of these conventional tumour markers is limited [8]. In such cases, imaging remains the only diagnostic tool; therefore, new tumour markers with a better performance are needed to tailor diagnosis and treatment.

In recent years, the analysis of epigenetic modifications showed that non-coding microRNAs (miRNAs) could be promising novel biomarkers for different cancers.

MiRNAs are a new class of small non-coding RNA molecules, consisting of 18–24 nucleotides, which play an important role in the regulation of gene expression after transcription. They usually have an inhibiting action with suppression of transcription or destruction of messenger RNA (mRNA), and they can be found in all tissues of the body [11,12].

Some MiRNAs have been found to play a role in the genesis of tumours and are profusely expressed in neoplastic tissue and released from tumour cells in the blood, where they remain stable and can be measured by quantitative polymerase chain reaction (qPCR) and used as biomarkers [13].

A subset of MiRNAs, MiRNA-371a-3p (M371), is involved in regulation of embryonic stem cell differentiation and was detected in GCT tissue and in serum of GCT patients, showing a very high capability to discriminate GCT patients from healthy subjects [14].

Previous studies have already demonstrated that M371 outperforms conventional tumour markers and could help to simplify diagnosis, follow-up, and early detection of recurrences in GCT patients [8,9,10,11].

In a previous study conducted by our group in 2023, we already evaluated the use of M371 under real-life conditions in 96 assays, performed on patients both at initial diagnosis and during follow-up. However, the primary aim of that study was to establish the M371-test on the Rotor-Gene Q (Qiagen) thermocycler and to evaluate its technical performance on a platform different from the LightCycler 480 II (Roche Diagnostics), which was originally used by the manufacturer to develop the test [15].

Given the technical validation achieved in our previous work, the present study aimed to evaluate the diagnostic accuracy of the M371-test under routine clinical conditions, comparing its performance to the conventional markers AFP, β-HCG, and LDH in patients suspected of GCT. The tests were performed prospectively by routine hospital personnel in a real-world setting, demonstrating the reliability of the test in everyday practice and confirming the suitability of the previously validated thermocycler.

## 2. Materials and Methods

### 2.1. Patients

Ninety-four M371-tests were performed in 94 consecutive patients (median age 37 years; range 14–66 years) and included in this prospective study, conducted under real-life conditions at our institution between March 2021 and May 2025.

The study was conducted in accordance with the Declaration of Helsinki and approved by the Ethic Committee of the Provincial Hospital of Bolzano with the number 97-2021. All patients provided written informed consent.

The only exclusion criterion was refusal to participate in the study.

Seventy-five tests (80.2%) were performed in 75 consecutive patients with suspicion of GCT, while the remaining 19 M371-tests (19.8%) were performed in a control group of healthy subjects with negative testicular ultrasound but benign conditions of epididymis or spermatic cord; these subjects were aged 19–73 years (median age 49) and had no current urological comorbidities.

One test (1.05%) was considered invalid and excluded from the analysis due to the severe hemolysis of the sample. Another 2 patients were excluded from the final evaluation due to a final histological diagnosis of testicular lymphoma and testicular liposarcoma.

In total, 91 M371-tests (72 in patients with suspicion of GCT and 19 in healthy individuals) were evaluated and compared with AFP, β-HCG, and LDH, using histological diagnosis as gold standard for those suspected of GCT.

Secondary objectives included investigating the correlation between preoperative M371 levels and tumour size (defined according to the diameter reported by the pathologist in the final histology report) in non-metastatic patients, as well as assessing its correlation with clinical stage in the entire GCT cohort.

The control group of healthy individuals was also included to verify the reliability and specificity of the results in a non-pathological setting.

Following surgical procedures, all patients underwent regular follow-up at our outpatient department.

The characteristics of the patient population are summarized in Table 1.

A flowchart of the patient cohort is shown in Figure 1.

### 2.2. Blood Sampling

In our study, M371 expression was analyzed in blood serum samples, usually within 3 days from surgery. To reduce changes in miRNA levels, the blood samples should be delivered to the laboratory within 30 min after collection, immediately separated into serum fraction, and stored at very low temperatures: delays, elevated temperatures, and high RNAase activity in blood, as well as the short half-life of M371, can degrade the sample and compromise the performance of the test. To ensure reproducibility, the procedure was standardized: blood samples were obtained from a cubital vein in our department, immediately placed in a transport box in a vertical position, and hand-delivered by one of our healthcare assistants directly to the molecular laboratory of our hospital (5 minutes walk), where a specialized technician processed them according to the manufacturer’s guidelines [16].

Whole blood samples were collected in 9 mL Vacuette^®^ Cat serum separator clot activator tubes, left at room temperature for 10 min to 1 h to complete clotting and then processed to serum by centrifugation and stored in 1 mL aliquots at −80°, before further processing.

Serum levels of M371 were determined as described in our previous study [15]. In brief, a miRNeasy Mini Kit (Qiagen, Hilden, Germany) was used to extract total RNA from 100 μL of serum, according to the manufacturer’s instructions [16]. Relative quantification of serum miR-371a-3p was performed using the M371-test (mir|detect GmbH, Bremerhaven, Germany), based on real time-PCR. The serum levels of M371 were measured relative to an internal reference miRNA, using the real-time Thermocycler Rotor-Gene Q (Qiagen), following validation of the platform as reported previously [15].

The cut-off level for positivity was set at RQ 5, according to Dieckmann et al. [14].

The conventional serum tumour markers AFP, β-HCG, and LDH, were measured according to laboratory guidelines.

All study activities had been conducted according to the Declaration of Helsinki of the World Medical Association.

### 2.3. Statistical Analyses

Statistical analysis was performed using R software (version 4.4.1; R Foundation for Statistical Computing, Vienna, Austria).

Continuous variables were tested for normality using the Shapiro–Wilk test and were found not to follow a normal distribution; therefore, they are presented as median and interquartile range (IQR). Categorical variables are summarized as absolute counts and percentages.

The sensitivity, specificity, positive predictive value (PPV) and negative predictive value (NPV) of M371 were evaluated and calculated using histology as gold standard and a RQ = 5 as cut-off value. The choice of RQ = 5 was based on the large multicentric study by Dieckmann et al. [14] and on the manufacturer’s recommendation [16], which identified this threshold as the most reliable.

The diagnostic accuracy of M371 was assessed by receiver operating characteristic (ROC) curve analysis, using the same parameters. The area under the curve (AUC) was calculated to evaluate the ability of MiRNA-371 to discriminate GCTs from benign lesions or control cases. An optimal diagnostic threshold of RQ = 5.34 was then estimated using Youden’s index, which maximizes the sum of sensitivity and specificity. Sensitivity, specificity, PPV, and NPV were also calculated, based on this optimized cut-off value.

ROC curves were also generated for the traditional serum tumour markers AFP, β-HCG, and LDH, and their AUCs were calculated according to the available data, to allow comparison with M371 performance.

The correlation between M371 expression levels and tumour diameter was assessed using Spearman’s rank correlation coefficient, given the non-parametric distribution of the data. The association between levels of M371 and tumour stage was evaluated using the Mann–Whitney U test.

Raw Ct values were not available, as only validated RQ data were provided by the molecular laboratory for clinical use.

## 3. Results

In the control group, M371 was negative in all cases, as was β-HCG, yielding 100% specificity for both markers. AFP was negative in 16 out of 19 samples (84.2%) and LDH in 17 out of 19 samples (89.4%). As detailed in Table 1, control subjects had a median age of 49 years and no urological comorbidities.

Among the 72 cases with suspected GCT, 44 (61.1%) were confirmed by histology: 24 (54.4%) were pure seminomas and 20 (45.4%) were non-seminomas, including 1 case of pure teratoma.

The M371-test showed a sensitivity of 90.9% compared to 20.4% for AFP, 40.9% for LDH and 43.1% for β-HCG. Specificity was 89.3% for M371, 96.4% for both AFP and LDH, and 100% for β-HCG. The PPV was 93% for Mi371, 90.0% for AFP, 94.7% for LDH, and 100% for β-HCG, whereas the NPV was 86.2% for Mi371, 43.5% for AFP, 50.9% for LDH, and 52.8% for β-HCG.

Stratifying marker performance according to the tumour histology, M371 was positive in 21 out of 24 cases with seminomatous tumours, yielding a sensitivity of 87.5%, compared to 0% for AFP, 29.1% for LDH, and 33.3% for β-HCG. Among non-seminomatous tumours, M371 was positive in 19 out of 20 cases, showing a sensitivity of 95.0% compared to that of 45.0% for AFP and 55.0% for both LDH and β-HCG. Notably, excluding the single case of pure teratoma, which was negative as expected, the sensitivity of M371 in this subgroup would have reached 100% (Table 2).

Receiver operating characteristic (ROC) curve analysis showed an area under the curve (AUC) of 0.92 for M371, compared with 0.615 for AFP, 0.721 for LDH, and 0.833 for beta-HCG (Figure 2).

When considering RQ ≥ 5.34 as a positive threshold, as suggested from the ROC curve, we observed only a slight decrease in sensitivity (from 90.9% to 89.0%) and NPV (from 86.2% to 84.0%), with an appreciable improvement in specificity (from 89.3% to 93%) and PPV (from 93.0% to 95.0%).

Of the 72 tested cases with suspicion of GCT, M371 was positive in 44 cases and negative in 28 cases, with four false negative and three false positive results. Among the four false negative cases, one was a pure teratoma and the remaining three were pure seminomas. All four showed undetectable M371 expression with a RQ value equal to 0, despite tumour sizes of 12, 15, and 32 mm in the seminomas. The three false positive results were found in patients with benign conditions (one area of sclerosis, one simple cyst, and one cyst with hemorrhagic infiltration), showing RQ values of 5.1, 7.8, and 104. Two of these cases occurred at the beginning of the test utilization at our site: in 1 of these cases, the test was later repeated, yielding a negative result.

The diameter of the tumours ranged from 6 to 100 mm (median 30 mm). Among the 31 non-metastatic patients (31/44: 70.4%), and therefore classified as clinical stage Ia/Ib, which also comprised the three false negative seminomas and the teratoma, a strong correlation was observed between expression of M371 and tumour size, with a Spearman correlation coefficient of 0.74 (Figure 3).

All M371-positive tumours < 20 mm (5/31: 16.1%) showed an RQ value < 20, while all tumours ≥ 20 mm had RQ value ≥ 20. Of note, in all lesions > 50 mm (8/31: 25.8%) the preoperative RQ value of M371 exceeded 400.

Furthermore, M371 levels were significantly associated with tumour stage. Patients with non-metastatic disease (Stage Ia/Ib; n = 31) showed a median M371 level of 98.36 (IQR: 15.20–784.24), whereas those with metastatic disease (Stage Is, II, III; n = 13) exhibited a markedly higher median level of 1128.35 (IQR: 313.00–1910.00). This difference was statistically significant (*p* = 0.015), indicating a strong association between increasing M371 expression and greater tumour burden (Table 3).

When analyzing marker performance by tumour stage individually—including stage Is, stage II (subgroups a, b, c), and stage III (subgroups a and b)—M371 showed higher sensitivity compared to conventional serum markers AFP, LDH, and β-HCG. The only exceptions were Stage Is and Stage IIIb, where all markers, including M371, showed sensitivity of 100%. A detailed comparison of sensitivities of the markers across tumour stages is reported in Table 3.

## 4. Discussion

Several studies reported a significant increase in the level of microRNA-371a-3p in patients with GCT [12,15,17,18], since Murray et al. in 2011 [18] proposed it for the first time as a novel marker for testicular cancer. In the multicenter study from Dieckmann et al. [17], the M371-test achieved a sensitivity of 90.1% and a specificity of 94% for primary diagnosis of GCT (with exception for pure teratoma, which does not express the marker). Two further prospective studies, by Nappi in 2019 with 132 cases [19], and by Myklebust in 2021 with 180 cases [20], also concerning primary diagnosis, reported a sensitivity of 96% and 89%, respectively, and a specificity of 100% in both studies.

Our previous prospective study from 2023 showed a significantly lower overall sensitivity of 73.7% and a specificity of 75% [15]. This discrepancy was likely due to the fact that our previous study was also conducted under real-life conditions, during an early phase when the entire process, from blood collection to laboratory processing, still needed to be standardized. Moreover, for the first time, a different real-time platform was used, the Rotor-Gene Q real-time thermocycler (Qiagen), which required a learning curve. In our current study, which includes only patients at primary diagnosis, we observed a significant improvement in the performance of M371, with an overall sensitivity of 90.9% and specificity of 89.3%. Our findings are in line with those reported in the other studies [14,17,19], confirming the excellent performance of this marker, even when applied under real-life conditions and with a different thermocycler.

However, despite these excellent results, some differences in specificity and sensitivity still remain.

Among our three false positive cases (one area of sclerosis, one simple cyst, and one cyst with hemorrhagic infiltration), two of them—those with RQ of 7.8 and 104—occurred at the very beginning of our study, during the initial learning curve with the different thermocycler [15]. These early cases may have been affected by technical inaccuracies in the analyses and influenced the final M371 value, contributing to our slightly lower specificity. In one of these cases, the patient with RQ value of 104, the test was repeated later and yielded a negative result, suggesting that the initial positive result may have been false [21], although RNA degradation during storage of the sample at −80 °C cannot be ruled out. The third false positive case, which showed only a very slight elevation (RQ = 5.1), occurred somewhat later, but still more than 2 years ago: it could still be explained by a technical problem or, alternatively, it is possible that a small subset of patients may have M371 levels slightly above the commonly accepted threshold [14], even in the absence of disease.

This overall interpretation is supported by the fact that no additional false positive results have been observed in the last two years of continuous testing. Furthermore, all samples from our control group have consistently shown negative M371 values, increasing the reliability of the test and its specificity [17,21,22]. In addition, using the optimized cut-off of RQ = 5.34 would have resulted in zero false positives in our cohort over the last 36 months.

Regarding false negative results, one case involved a pure teratoma, a tumour subtype, which does not express M371, and was therefore expected to be negative. Consistent with previous reports [23,24], this lack of expression may reflect the nature of teratomas, which are composed of differentiated tissues similar to normal somatic morphology and therefore do not produce specific circulating biomarkers. Of note, the other three cases were histologically confirmed pure seminomas, which showed a complete absence of M371 expression (RQ = 0), despite the tumour size more than 1 cm, with one exceeding 3 cm in diameter [23].

Two of these cases occurred at the beginning of our experience: in both of them, we cannot rule out potential problems before the analysis, like some delays of the transportation of blood samples to the molecular laboratory or exposure to high temperatures, which may have affected the RNA degradation and contributed to undetectable M371 [21]. The third case occurred later in our study, and no apparent technical issues were identified. However, in this case, as well as in one of the previous two, the histological examination showed marked fibrosis and impaired spermatogenesis in all testicular tissue. These features could be interpreted as a potential factor affecting the level of M371 expression [21,22]. However, another explanation about the false negative results could be, as reported from Dieckmann et al. [14], a higher degree of differentiation of the tumoral tissue of these patients, with morphologic similarity to spermatogonia, and therefore a lower miR-371a-3p expression. However, extensive tissue necrosis within the tumour could also contributing to reduced detectable M371 levels.

These observations suggest that, even if M371 is a very highly sensitive marker for GCT, some false negative results can occasionally occur, not only in teratomas but also in pure seminomas. Summarizing, potential contributing factors are technical or handling issues, extensive necrosis or fibrosis of the tissue, compromised testicular environment, or high degree of tumour cells differentiation [14,21,22,23].

Our findings underline the relevance of technical accuracy, with standardization of sample collection and processing procedures, and accumulated laboratory experience to ensure the reliability of the results. Further studies are needed to understand the influence of some histological features, like fibrosis, low cellularity, or germ cell aplasia, on the expression of M371.

Nevertheless, in our cohort, if we exclude the case with teratoma and the other two additional potential false positive and false negative results observed during the initial learning curve, the sensitivity of M371 in our study rises to 97.7% and the specificity to 96.4%. Moreover, even if we do not exclude these cases, by using the optimized cut-off value, derived from the ROC-curve analysis, we were still able to improve our specificity (from 89.3% to 93.0%) and PPV (from 93.0% to 95.0%).

Of note we also observed a strong positive correlation between expression level of M371 and tumour diameter in non-metastatic patients, as demonstrated by Spearman’s correlation coefficient of 0.74, indicating that M371 may reflect tumour burden in early-stage disease. Lower RQ values were associated with smaller lesions, as also reported from Dieckmann et al. [17] and Nazzani et al. [25], and confirmed in our study, where all M371 positive lesions < 20 mm had a RQ value < 20, supporting its potential utility as diagnostic marker, also in cases of limited tumour volume. Nevertheless, in our cohort, M371 failed to detect two seminomas in this size range, in addition to the teratoma, indicating that a negative result should be interpreted with caution. This is consistent with the findings of Nazzani et al. [25], which also highlighted the risk of missed detection in small testicular masses.

Due to the limited number of small tumours in our cohort, we did not perform a stratified sensitivity analysis by tumour size.

In contrast to the findings reported in other studies [17,26,27], no significant differences were observed between seminomas and non-seminomas in our cohort; however, the limited number of patients in each subgroup precludes any definitive conclusions.

Furthermore, M371 levels were significantly correlated with tumour stage. Patients with clinical stage Ia/Ib showed markedly lower M371 levels if compared to those with advanced stages, further supporting a potential correlation between M371 expression and tumour burden. In particular, persistently elevated M371 levels after orchiectomy or chemotherapy might indicate the presence of residual disease or micro-metastasis and could therefore assist in the treatment planning and clinical decision-making. Interestingly, stratifying the patients by stage, M371 maintained a very high sensitivity across all stages, outperforming the conventional markers, except in stage Is and IIIb, where all markers were positive.

In summary, M371 demonstrated an excellent performance even in real life conditions, outperforming the conventional tumour markers in terms of sensitivity and NPV, without losing specificity and PPV, even in small lesions. Furthermore, the test remained highly sensitive at all stages, confirming its diagnostic reliability across the disease spectrum.

We acknowledge the single-center character of the study and the relative limited number of cases, particularly for the subgroup analysis. Additionally, the initial learning curve may have influenced the early phase of the study, with a potential effect on overall results.

Moreover, the clinical use of M371 is still limited, due to the technical problems related to the sample handling, storage, and analysis, with the need of a molecular laboratory and experienced pathologist and technicians. In addition, the costs of M371 are still relatively high and variable across healthcare systems.

Notably, although M371 has shown excellent diagnostic performance, its lack of sensitivity for teratoma, and rarely for seminomas, may currently limit its ability to safely rule out malignancy. Therefore, in cases of suspected testicular lesions, inguinal exploration remains necessary. Nevertheless, in selected cases with small, indeterminate lesions and negative M371, a more conservative approach, such as active surveillance or testis-sparing surgery, could be considered.

Further multicentric studies are needed to better define the role of M371, especially in treatment planning and during follow-up [26]. At present, M371 appears a promising diagnostic tool to support the clinical decision-making, but broader validation is still required before routine implementation.

However, despite these limitations, M371 still outperforms conventional serum markers in terms of sensitivity and overall diagnostic accuracy, confirming its potential as a reliable diagnostic tool in daily clinical practice.

Nonetheless, it should be noted that this is a real-life study and, to our knowledge, it is currently the only one that assesses the performance of M371 under routine clinical conditions, also confirming the suitability of the previously validated thermocycler. This aspect can add relevance to our findings, despite the limitations described above.

## 5. Conclusions

Under real life conditions, the M371-test shows a significantly better performance compared to the conventional markers, even in small lesions. Despite all potential limitations and technical issues, M371 appears to be a reliable diagnostic tool in routine clinical practice to improve the management of the patients with GCT in clinical decision-making and treatment planning. Larger multicentric prospective studies and standardized pre-analytical procedures are still required to validate its role and eventually to replace or reduce the need for imaging in selected settings.

## Figures and Tables

**Figure 1 pathophysiology-32-00054-f001:**
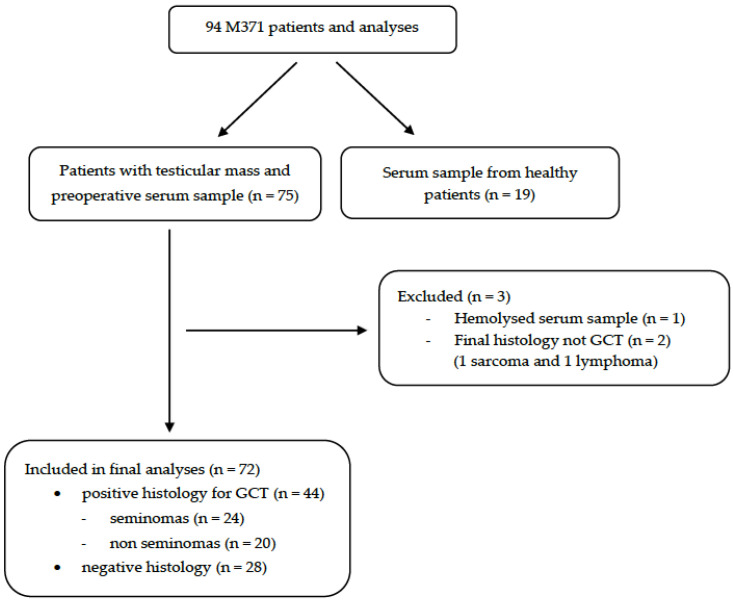
Study flowchart. Out of 75 patients with testicular mass and available preoperative serum, 3 were excluded: 2 due to final histology not confirming germ cell tumour (GCT), and 1 due to hemolyzed serum sample. The final study cohort included 72 patients.

**Figure 2 pathophysiology-32-00054-f002:**
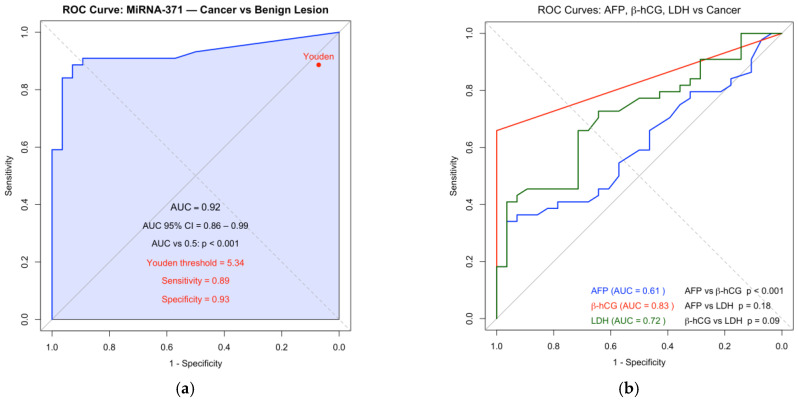
ROC curves with AUC of miRNA-371 (**a**) and of AFP, β-HCG, and LDH (**b**), including statistical comparisons between AUCs. Statistical comparison of AUCs showed that β-HCG significantly outperformed AFP (*p* < 0.001), while differences with LDH were not significant.

**Figure 3 pathophysiology-32-00054-f003:**
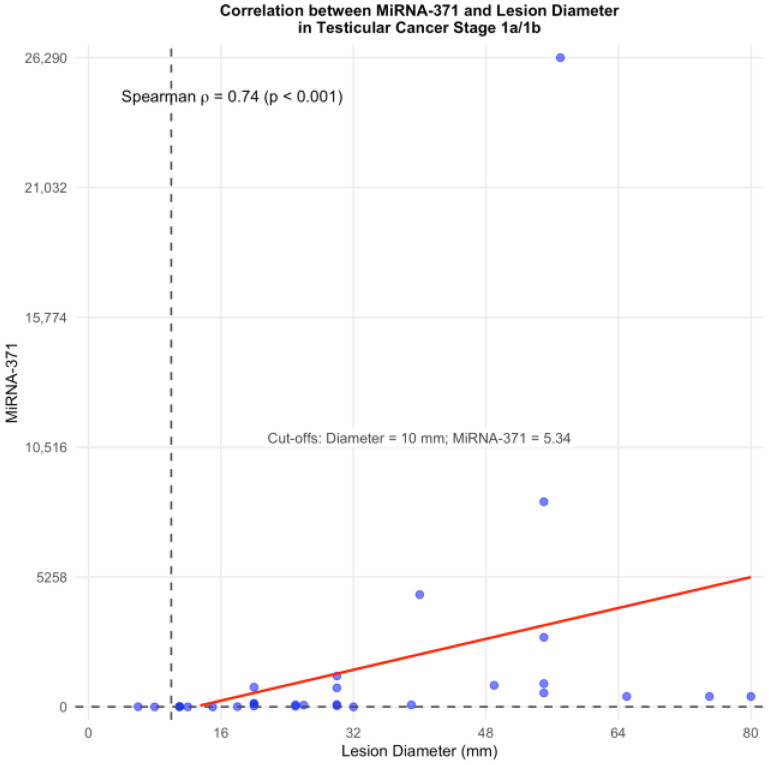
Spearman correlation between M371 and diameter in non-metastatic GCT. Individual data points are shown in blue, with the red line indicating the linear regression. The dashed vertical line marks the clinical cut-off diameter of 10 mm, while the annotation indicates the best cut-off value of M371 (5.34).

**Table 1 pathophysiology-32-00054-t001:** Baseline characteristics of the patient population.

**Total Patients and Analyses**	**94**
Age	range 14–84; median 38
Non-diagnostic Analyses	1
Excluded (no GCT histology)	2
Evaluable Analyses	**91**
- Control group	19 (age range 19–73; median 49)
- Suspicion of GCT on ultrasound	72 (age range 14–66; median 37)
Positive for GCT	44
Negative for GCT	28
**Tumour histology**	
- Seminomas	24
- Non-seminomas	20
mixed GCT	14
embryonal carcinoma	5
teratoma	1
**Tumour diameter (mm)**	Range 6–100; median 30
**pT stage**	
- pT1	20
- pT2	17
- pT3	5
- Extragonadal	1
- Burned out	1
**Clinical Stage (CS)**	
Non-metastatic (stage Ia/Ib)	**31** (Ia: 16; Ib: 15)
Metastatic (stage Is/II/III)	**13** (Is: 1; II: 6; III: 6)
Prognosis (IGCCCG)	good: 11; poor: 2

GCT: germ cell tumours; IGCCCG: International Germ Cell Cancer Cooperative Group.

**Table 2 pathophysiology-32-00054-t002:** Sensitivity, specificity, predictive values, and stratified sensitivity of M371, AFP, LDH, and β-HCG in patients with suspicion of GCT.

	Sensitivity (%)	Specificity (%)	PPV (%)	NPV (%)	Seminomas (%)	Non- Seminomas (%)	Non- Seminomas (Excl. Pure Teratoma) (%)
M371	90.9%	89.3%	93.0%	86.2%	87.5%	95.0%	100%
AFP	20.4%	96.4%	90.0%	43.5%	0%	45.0%	42.1%
LDH	40.9%	96.4%	94.7%	50.9%	29.1%	55.0%	57.8%
β-HCG	43.1%	100%	100%	52.8%	33.3%	55.0%	57.8%

M371: Micro-RNA 371; AFP: alpha-fetoprotein; LDH: lactate-dehydrogenase; β-HCG: beta-human chorionic gonadotropin.

**Table 3 pathophysiology-32-00054-t003:** Distribution of M371 levels and sensitivity of conventional serum markers according to tumour stage. Median M371 levels were significantly higher in metastatic compared with non-metastatic disease (*p* = 0.015).

Tumour Clinical Stage	N. Patients	Median M371 (IQR)	M371 Positivity (%)	AFP Positivity (%)	LDH Positivity (%)	β-HCG Positivity (%)	*p*-Value
1a/1b	31	98.36 (15.20–784.24)	100	-	-	-	0.015
Other tumour stages	13	1128.35 (313.00–1910.00)	100	-	-	-	
1s	1		100%	100%	100%	100%	
2a	2		100%	0%	100%	0%	
2b	2		100%	0%	50%	50%	
2c	2		100%	0%	50%	0%	
3a	4		100%	50%	100%	75%	
3b	2		100%	100%	100%	100%	

M371: Micro-RNA 371; AFP: alpha-fetoprotein; LDH: lactate-dehydrogenase; β-HCG: beta-human chorionic gonadotropin; IQR: interquartile range.

## Data Availability

The data of the present study are available on request from the corresponding authors (C.D.; E.T.).

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
