# Peer review of "MicroRNA-371a-3p Represents a Novel and Effective Diagnostic Marker for Testicular Germ Cell Tumours: A Real-World Prospective Comparison with Conventional Approaches"

_pathophysiology, 2025, doi:10.3390/pathophysiology32040054_

Round 1
Reviewer 1 Report
Comments and Suggestions for Authors
I review the manuscript “MicroRNA-371a-3p Represents a Novel and Effective Diagnostic Marker for Testicular Germ Cell Tumors: A Real-World Prospective Comparison with Conventional Approaches”by Palermo et al., which I consider addresses an important clinical need: the lack of reliable, sensitive serum markers for testicular germ cell tumors, especially in early-stage disease. The study evaluates miR-371a-3p (M371) as a diagnostic tool under real-life clinical conditions and compares its performance to traditional biomarkers (AFP, β-HCG, LDH). The work is clinically relevant, given the ongoing debate about whether M371 should be incorporated into diagnostic guidelines. The manuscript is well-structured, but several conceptual, methodological, and interpretative aspects require attention to strengthen its impact.,
Major observations
-While the ms demonstrates the superior diagnostic performance of M371, it does not sufficiently discuss the underlying pathophysiology of why M371 is expressed so robustly in seminomas and non-seminomas, but not in teratomas. I consider that the mechanistic framework is essential to understand limitations in pottential biomarker application. The discussion also would benefit from integrating hypotheses about cellular origin, differentiation state, and epigenetic regulation of miR-371a-3p expression. Without this, the clinical results are presented somewhat in isolation from biological rationale.
-The authors acknowledge the single-center and limited sample size, but the implications of these limitations are underexplored. Let me explain.,the control group is small (n=19) and not fully matched by age, comorbidities, or potential confounders. This undermines specificity claims. Exclusion of rare histologies (lymphoma, liposarcoma) is logical, but the decision to report their M371 values in discussion without clear interpretation creates confusion. False positives and negatives are attributed largely to “learning curve” issues. While plausible, this explanation seems weak without systematic analysis of pre-analytical variables. The reliance on retrospective interpretation raises concerns about reproducibility.
-The statistical analysis is adequate but lacks nuance. For instance:the ROC curve analysis identifies an “optimized” cut-off (RQ 5.34), yet the authors still base most of their discussion on the fixed manufacturer’s cut-off (RQ 5). A more transparent justification for choosing thresholds is needed.
-The correlation between tumor size and M371 expression (p=0.74) is strong, but the manuscript fails to discuss whether this could bias early-stage detection—small tumors may still be missed. A stratified sensitivity analysis by tumor size would help.
-Survival or prognostic outcomes are not reported. Even if beyond scope of their ms, a mention of ongoing follow-up would increase relevance.
-Fig 1 (flowchart) is clear but I consider simplistic; additional annotation on excluded cases would improve clarity. Fig3 (correlation plot) is informative, but the axes and cut-off lines should be standardized. ROC curves (Fig 2) could include statistical comparison between AUCs.
-Tables 2–5 are essential but presented with redundancy. Combining sensitivity/specificity across histology and stage into a single comprehensive table would improve readability.
-The reference list is solid, but several important recent works are missing:
-https://doi.org/10.3390/ijms25042156 discusses teratoma challenges more deeply in relation to false negatives.
-Multicenter validation studies with digital-droplet PCR (Myklebust et al. 2021, Sci Rep, cite 20) are mentioned but not critically analyzed; these methods may overcome some of the technical issues reported here.
-Nazzani et al. (2025) about S1STeM 371 trial is cited (cite 24)but under-discussed; its findings on small testicular masses should be contrasted with this cohort’s results.
-The conclusion strongly suggests M371 should be adopted in guidelines. However, the manuscript does not address important aspects to valídate this state like cost-effectiveness compared to imaging or surgery, feasibility in non-tertiary centers without advanced molecular labs and how clinicians should interpret discordant results ( M371 negative, imaging suspicious??).
Minor
This overstates clinical readiness and risks misleading readers about applicability.
Expand discussion on mechanisms of M371 expression and why teratomas and some seminomas fail to express it. Consider epigenetic or differentiation-related hypotheses.
Authors should provide stratified sensitivity/specificity by tumor size, subgroup analysis excluding “learning curve” cases, and comparative statistics between AUCs of different markers.
Reframe conclusions to emphasize promise rather than immediate guideline adoption, highlighting the need for multicenter, larger prospective validations and standardized pre-analytical handling.
Author Response
Thank you very much for taking the time to review this manuscript. Please find the detailed responses below with the corresponding revisions and corrections highlighted in the re-submitted files.
I review the manuscript “MicroRNA-371a-3p Represents a Novel and Effective Diagnostic Marker for Testicular Germ Cell Tumors: A Real-World Prospective Comparison with Conventional Approaches”by Palermo et al., which I consider addresses an important clinical need: the lack of reliable, sensitive serum markers for testicular germ cell tumors, especially in early-stage disease. The study evaluates miR-371a-3p (M371) as a diagnostic tool under real-life clinical conditions and compares its performance to traditional biomarkers (AFP, β-HCG, LDH). The work is clinically relevant, given the ongoing debate about whether M371 should be incorporated into diagnostic guidelines. The manuscript is well-structured, but several conceptual, methodological, and interpretative aspects require attention to strengthen its impact.
Major observations
-While the ms demonstrates the superior diagnostic performance of M371, it does not sufficiently discuss the underlying pathophysiology of why M371 is expressed so robustly in seminomas and non-seminomas, but not in teratomas. I consider that the mechanistic framework is essential to understand limitations in pottential biomarker application. The discussion also would benefit from integrating hypotheses about cellular origin, differentiation state, and epigenetic regulation of miR-371a-3p expression. Without this, the clinical results are presented somewhat in isolation from biological rationale.
We thank the reviewer for the valuable suggestion. As suggested, we have integrated the discussion with recent literature, addressing the biological rationale (line 374 until 377). We have added that M371shows no diagnostic utility in teratomas, likely because they are composed of well-or partially differentiated tissues, which mimic a normal somatic morphology and for this reason they lack to produce circulating biomarkers: Yodkhunnatham et al 2024 [23], and Partin et al 2021 [24]. Our clinical findings are consistent with this observation, but detailed studies on these mechanisms remain beyond the aim of our clinical analysis.
-The authors acknowledge the single-center and limited sample size, but the implications of these limitations are underexplored. Let me explain,the control group is small (n=19) and not fully matched by age, comorbidities, or potential confounders. This undermines specificity claims. Exclusion of rare histologies (lymphoma, liposarcoma) is logical, but the decision to report their M371 values in discussion without clear interpretation creates confusion. False positives and negatives are attributed largely to “learning curve” issues. While plausible, this explanation seems weak without systematic analysis of pre-analytical variables. The reliance on retrospective interpretation raises concerns about reproducibility.
We are grateful to the reviewers for their insightful comments. We acknowledge the limitations of the relatively small control group. In the revised manuscript, we now specify that the median age of the participants was 49 years and clarify that none of the patients had current urological comorbidities that could represent potential confounding factors (lines 126 and 127, 246 and 247). We changed also Table 1, adding these informations.
Regarding the inclusion of rare histological subtype, we respect the reviewers’ judgment and, should it be considered more appropriate for the clarity of the manuscript, we remove these data from the manuscript (lines 330 and 331).
With regard to the false positive tests:
the 3 false positive cases were found in patients with benign conditions (one area of sclerosis, one simple cyst and one cyst with hemorrhagic infiltration), with RQ values of 5.1, 7.8, and 104.
- In the case with RQ 104, a later repeated test, yielded a negative result, suggesting the possibility that the initial finding may have been false. However, as we cannot exclude RNA degradation during storage, a true false positive remains possible and for this reason, we included this result in our analysis as valid.
- One case showed only a minimal elevation (RQ=5.1). Based on our optimized cut-off of 5.34 (Youden index), this value would be within the negative range.
- Furthermore, it is also possible that a small subset of patients may have slightly elevated levels of M371 even in absence of specific disease
Concerning reproducibility, no additional false positive results have been observed in our laboratory over the past two years of continuous testing, which supports the robustness of our findings. Moreover, adopting our optimized cut-off of 5.34 would result in zero false positive over the last 36 months. We have added a phrase in the test (lines 371, 372 and 373)
We aknownledge the reviewers’ concerns and agree that future studies with larger control groups and extended follow-up warrented to further validation and reproducibility: we added a sentence at the end of Conclusions (lines 474, 475 and 476)
-The statistical analysis is adequate but lacks nuance. For instance:the ROC curve analysis identifies an “optimized” cut-off (RQ 5.34), yet the authors still base most of their discussion on the fixed manufacturer’s cut-off (RQ 5). A more transparent justification for choosing thresholds is needed.
The decision to adopt a cut-off value of 5 was guided by the multicenter study conducted by Dieckmann et al. (J Clin Oncol. 2019 Jun 1;37(16):1412-1423. doi: 10.1200/JCO.18.01480), which included 616 patients and identified RQ=5 as the optimal threshold according to the Youden index.
Furthermore, this RQ value is the one recommended by the manufacturer (https://www.mirdetect.de/cms-data/depot/deutsch/Gebrauchsanweisungen/Instructions-for-use-M371-Test_v13_EN.pdf; pag. 5). We therefore considered it (RQ=5) the most reliable and clinically applicable threshold for our study. We have added a clarification on this point in the text, lines 224, 225, and 226
At the same time, we acknowledge that in our prospective cohort we identified an optimized cut-off of 5.34 through ROC analysis. Naturally, this value could not have been known a priori: in the Results you can find a brief comparison showing how sensitivity and specificity would have changed if this threshold had been applied: lines 280, 281, 282, 283.
-The correlation between tumor size and M371 expression (p=0.74) is strong, but the manuscript fails to discuss whether this could bias early-stage detection—small tumors may still be missed. A stratified sensitivity analysis by tumor size would help.
We thank the reviewer for this important observation. In the revised discussion, from line 415 to line 423, we now report that in our cohort M371 was detectable in 5 out of 7 small tumours (<20 mm; 71.4%), supporting its role in early-stage disease. However, two false negative seminomas in this subgroup, in addition to the teratoma, indicate that a negative result should be interpreted with caution. This highlights that small malignant lesions may occasionally escape detection. Due to the limited number of small tumours, we did not perform a stratified sensitivity analysis, but we agree that such analyses in larger cohorts would be valuable to further clarify this aspect. In the result section we have highlighted that in non metastatic group were included also the teratoma and the 3 false negative seminomas (line 295) and that the 5 lesions <20 mm were the 5 M371 positive lesions (line 307).
-Survival or prognostic outcomes are not reported. Even if beyond scope of their ms, a mention of ongoing follow-up would increase relevance.
We thank the reviewer for this observation. We agree that survival and prognostic outcomes are clinically relevant, though they were beyond the scope of the present study. Follow-up data for this cohort are currently being collected but are not yet mature.
-Fig 1 (flowchart) is clear but I consider simplistic; additional annotation on excluded cases would improve clarity. Fig3 (correlation plot) is informative, but the axes and cut-off lines should be standardized. ROC curves (Fig 2) could include statistical comparison between AUCs.
We thank the reviewer for this useful suggestions.
Answer figure 1: Although only three patients were excluded, we have updated Figure 1 to clearly indicate the reasons for exclusion. Additional text has been included in the figure caption.
Answer Figure 3: in the revised Figure 3, we have standardized the axes and added a cut-off line in the correlation plot to improve clarity. Additional text has been included in the figure caption.
Answer Figure 2: we have added the statistical comparison between AUC as suggested. Additional text has been included in the figure caption
-Tables 2–5 are essential but presented with redundancy. Combining sensitivity/specificity across histology and stage into a single comprehensive table would improve readability.
We thank the reviewer for this suggestion. We have revised the tables: the previous Tables 2 and 3 have been combined in one table 2, reporting the overall diagnostic performance together with the stratificated results by histology. Also the table 4 and 5 have been combined in one table 3, which summarize the sensitivity of M371 and of the markers across the tumour stages. This reorganization, as you have suggested, avoids redundancy and improves readability without loose relevant information.
-The reference list is solid, but several important recent works are missing:
Thank you for your suggestion. In the manuscript we had already included the reference “Yodkhunnatham, N et al. MicroRNAs in Testicular Germ Cell Tumors: The Teratoma Challenge. Int. J. Mol. Sci. 2024, 25, 2156. https://doi.org/10.3390/ijms25042156” as reference number 23, in the discussion when addressing false negative results in teratomas. To stress this point, we added Partin et al, Campbell-Walsh-Wein Urology, 2021, as reference 24, to highlights the similar morphology of teratomas to normal tissue which could explain the lack of M371 (from line 375 to line 378).
-Multicenter validation studies with digital-droplet PCR (Myklebust et al. 2021, Sci Rep, cite 20) are mentioned but not critically analyzed; these methods may overcome some of the technical issues reported here.
We thank the reviewer for this valuable point. But we cited Myklebust et al. to highlight the high sensitivity of M371. We agree that the role of different PCR techniques is of interest, but a detailed methodological comparison was beyond the scope of our study.
-Nazzani et al. (2025) about S1STeM 371 trial is cited (cite 24) but under-discussed; its findings on small testicular masses should be contrasted with this cohort’s results.
We appreciate this suggestion. In the revised discussion, we now contrast our findings with those of Nazzani et al. Specifically, while our results indicate that M371 can identify most small tumours, with 71.4% positivity in lesions <20 mm, without counting the teratoma, we also observed two false negative seminomas, underscoring the limitations of the test in this subgroup. This is consistent with Nazzani et al., which similarly emphasized the risk of missed detection in small testicular masses. We added 2 sentences from line 417 to line 421.
-The conclusion strongly suggests M371 should be adopted in guidelines. However, the manuscript does not address important aspects to valídate this state like cost-effectiveness compared to imaging or surgery, feasibility in non-tertiary centers without advanced molecular labs and how clinicians should interpret discordant results (M371 negative, imaging suspicious??).
We thank the reviewer for these important considerations regarding the implementation of M371 testing in clinical practice. We acknowledge that while M371 shows promising diagnostic accuracy and potential to help stratify patient risk and guide treatment planning, several practical aspects still need to be addressed before its routine adoption. The cost-effectiveness of M371 compared to imaging or surgery varies across healthcare systems and remains to be fully evaluated. Additionally, the test currently requires specialized molecular laboratories and experienced personnel, limiting its availability primarily to tertiary centers. In the discussion we have already highlighted the need of a specialized laboratory (lines 445, 446, 447) and the problems about false negative results (from line 449, to line 454). Moreover, this paper is about diagnosis of GCT, therefore it’s not the aim of the study to discuss about suspicious imaging with negative M371, as after orchifunicolectomy. About the costs, they remain variable across countries, making precise cost-effectiveness estimates difficult. However, we added a line in the manuscript to address this last point (line 447 and 448).
Minor
This overstates clinical readiness and risks misleading readers about applicability.
We thank the reviewer for this valuable observation. We understand the concern that some expressions in the manuscript might give an impression of clinical readiness that overstates the current applicability of M371 testing. We have added lines 456, 457 and 458 in the discussion section to emphasize that, despite its promising diagnostic performance, M371 remains an investigational tool, whose routine clinical use is currently limited.
Expand discussion on mechanisms of M371 expression and why teratomas and some seminomas fail to express it. Consider epigenetic or differentiation-related hypotheses.
We appreciate this suggestion. However, although we agree that epigenetic and differentiation-related mechanisms may contribute to the lack of M371 expression in some cases, our study was designed as a real-life clinical analysis and we limited our discussion to histological and morphological aspects supported by current clinical evidence. A deeper exploration of epigenetic mechanisms would go beyond the scope of this manuscript.
Authors should provide stratified sensitivity/specificity by tumor size, subgroup analysis excluding “learning curve” cases, and comparative statistics between AUCs of different markers.
We appreciate this suggestion. However, since our study is a prospective real-life analysis, we preferred to include all cases, including the “learning curve” cases, to avoid retrospective bias. Tumour size correlation was reported and AUC comparison between markers are presented in figure 2, as already request.
Reframe conclusions to emphasize promise rather than immediate guideline adoption, highlighting the need for multicenter, larger prospective validations and standardized pre-analytical handling.
We thanks the reviewer for this suggestion. We have revised the conclusion with more cautious words and added the need to of larger multicentric studies and standardized pre-analytical procedures (lines 474, 475, 476).
Reviewer 2 Report
Comments and Suggestions for Authors
This study addresses the clinical diagnostic value of miR-371a-3p in testicular germ cell tumors and provides meaningful real-world evidence. The results are promising and align with prior research, supporting the potential of miR-371a-3p as a biomarker. However, the manuscript requires significant improvement in methodological clarity, technical detail, and data presentation to strengthen reproducibility and highlight innovation. More comprehensive cohort information and transparent analytical methods are also needed before the work can be considered for publication.
Comments:
- I recommend that the authors provide additional information on miR-371a-3p qPCR Ct values in a supplementary table. It is important to clarify whether the overall relative expression follows a normal distribution. A histogram and/or density plot would be useful. If the distribution is non-normal or near-normal, non-parametric tests (i.e., Mann–Whitney/Wilcoxon) should replace parametric tests (i.e., Student’s t-test). Please verify.
- The Methods section is not detailed enough for proper assessment. In particular, information on the patient cohort is missing. The selection and inclusion criteria should be clearly stated, and additional clinicopathological data should be included, such as stage, grade, lymph node status, and LVI status.
- Since many studies have already reported miR-371a-3p as a novel serum biomarker in testicular germ cell tumors, the authors should emphasize the innovative aspects of this study more clearly in the Introduction and Discussion sections.
- It is well known that liquid biopsy faces clinical limitations, including (1) low biomarker concentration, limiting detection sensitivity, and (2) technical challenges in separation and enrichment, making results highly dependent on operator expertise. The authors should provide more detailed technical information on sample handling and quality control (e.g., processing time, transport conditions, temperature monitoring) from collection to qPCR. They should also discuss how such errors can be minimized in routine practice to ensure reproducibility of miR-371a-3p as a biomarker.
Author Response
Thank you very much for taking the time to review this manuscript. Please find the detailed responses below with the corresponding revisions and corrections highlighted in the re-submitted files.
This study addresses the clinical diagnostic value of miR-371a-3p in testicular germ cell tumors and provides meaningful real-world evidence. The results are promising and align with prior research, supporting the potential of miR-371a-3p as a biomarker. However, the manuscript requires significant improvement in methodological clarity, technical detail, and data presentation to strengthen reproducibility and highlight innovation. More comprehensive cohort information and transparent analytical methods are also needed before the work can be considered for publication.
Comments:
1. I recommend that the authors provide additional information on miR-371a-3p qPCR Ct values in a supplementary table. It is important to clarify whether the overall relative expression follows a normal distribution. A histogram and/or density plot would be useful. If the distribution is non-normal or near-normal, non-parametric tests (i.e., Mann–Whitney/Wilcoxon) should replace parametric tests (i.e., Student’s t-test). Please verify.
We thank the reviewer for this comment. However, the raw qPCR Ct data were not available to us, as the molecular laboratory performing the test provided only validated RQ values for clinical use. For trasparency we have clarified this in the Statistical Analysis section of the manuscript (lines 240, 241). If required, we may ask the laboratory whether Ct values can be made available, but we believe their inclusion would not affect the clinical validity of our results.
As specified in the Statistical Analysis section, we already tested all continuous vaiables for normality using the Shapiro–Wilk test, which confirmed that RQ values did not follow a normal distribution. Accordingly, we applied only non-parametric tests (Mann-Whitney U test, Spearman’s rank correlation). We therefore believe that our statistical approach adequately addresses the distributional characteristics of the data.
2. The Methods section is not detailed enough for proper assessment. In particular, information on the patient cohort is missing. The selection and inclusion criteria should be clearly stated, and additional clinicopathological data should be included, such as stage, grade, lymph node status, and LVI status.
We thank the reviewer for the comment. We would like to clarify that our study included all consecutive patients presenting with suspected testicular neoplasia without any selection as already described in line 116; to clarify it, as requested, we also added the words: “75 consecutive” after “were performed in” in line 128. We removed the word “consecutive” in line 123. The only exclusion criterion was refusal to participate in the study and we added this sentence in the text, line 122. The control group consisted of patients with benign lesions of the spermatic cord or epididymis, with no current urological comorbidities: we added this phrases and the age on line 126, 127. The table 1 was updated as requested with age of all groups of patients.
We agree that detailed clinicopathological data are important. All available relevant data, including age, clinical stage and lymph node status (defined by clinical stage) are reported in Table 1. We missed to report the pT stage, which also defines the lymphovascular invasion, and the histological type, which were added in Table 1. In testicular cancer, tumor grade is not a parameter and therefore was not included. We hope this clarifies the patient selection and cohort characteristics.
3. Since many studies have already reported miR-371a-3p as a novel serum biomarker in testicular germ cell tumors, the authors should emphasize the innovative aspects of this study more clearly in the Introduction and Discussion sections.
We thank the reviewer for the comment. We have added a sentence in the introduction (lines 109, 110, 111) emphasizing that the tests were performed prospectively by routine hospital personnel in a real-world setting. This addition highlights the reliability of the M371 test in everyday practice and confirms the suitability of the previously validated thermocycler, further supporting the relevance and novelty of our study. Moreover we believe that the last paragraph of the discussion (from line 462 to line 465) already highlight the relevance and novelty of our study, emphasizing the real-life assessment of M371 performance, its superiority over conventional markers, and its potential clinical implications. Adding further statements might be redundant, but we have add a concise sentence to highlight also the use of a different previously validated thermocycler (line 464).
4. It is well known that liquid biopsy faces clinical limitations, including (1) low biomarker concentration, limiting detection sensitivity, and (2) technical challenges in separation and enrichment, making results highly dependent on operator expertise. The authors should provide more detailed technical information on sample handling and quality control (e.g., processing time, transport conditions, temperature monitoring) from collection to qPCR. They should also discuss how such errors can be minimized in routine practice to ensure reproducibility of miR-371a-3p as a biomarker.
We thank the reviewer for this important point. In the revised manuscript we have expanded the Materials and Methods section to better describe our standardized procedure for sample handling and transport, lines 188 and 189 and from line 193 to 198. Specifically, blood samples were collected in our department, immediately placed in a vertical position inside a transport box, and hand-delivered by a healthcare assistant directly to the molecular laboratory, where a specialized technician processed them according to the manufacturer’s guidelines. This workflow was designed to minimize pre-analytical variability and ensure reproducibility of the M371 test.
Round 2
Reviewer 1 Report
Comments and Suggestions for Authors
I have reviewed the authors' new version of their ms and have improved it considerably. I also believe they have adequately addressed the questions raised. This version may be considered for publication due to its contribution to miRNAs as diagnostic markers in TGCT.